# Neural Fine-Tuning Search for Few-Shot Learning

**Panagiotis Eustratiadis[†], Łukasz Dudziak[‡], Da Li[‡], Timothy Hospedales[†‡]**
[†]University of Edinburgh
[‡]Samsung AI Center, Cambridge

## Abstract

In few-shot recognition, a classifier that has been trained on one set of classes is required to rapidly adapt and generalize to a disjoint, novel set of classes. To that end, recent studies have shown the efficacy of fine-tuning with carefully-crafted adaptation architectures. However this raises the question of: How can one design the optimal adaptation strategy? In this paper, we study this question through the lens of neural architecture search (NAS). Given a pre-trained neural network, our algorithm discovers the optimal arrangement of adapters, which layers to keep frozen, and which to fine-tune. We demonstrate the generality of our NAS method by applying it to both residual networks and vision transformers and report state-of-the-art performance on Meta-Dataset and Meta-Album.

## 1 Introduction

Few-shot recognition (Lake et al., 2011; Miller et al., 2000; Wang et al., 2020b) aims to learn novel concepts from few examples, often by rapid adaptation of a model trained on a disjoint set of labels. Many solutions adopt a meta-learning perspective (Finn et al., 2017; Lee et al., 2019; Ravi & Larochelle, 2017; Snell et al., 2017), or train a powerful feature extractor on the source classes (Tian et al., 2020; Wang et al., 2019) – both of which assume that the training and testing classes are drawn from the same underlying distribution e.g., written characters (Lake et al., 2015), or ImageNet categories (Vinyals et al., 2016). Later work considers a more realistic and challenging setting of few-shot adaptation not only across visual categories, but also across diverse visual domains (Triantafillou et al., 2020; Ullah et al., 2022). In this cross-domain problem variant, customising the feature extractor for novel domains is important, and several studies address this through dynamic feature extractors (Bateni et al., 2020; Requeima et al., 2019) or ensembles of features (Dvornik et al., 2020a; Li et al., 2021; Liu et al., 2021a). Another group of studies employ heuristically-motivated fine-tuning strategies for adaptation (Dhillon et al., 2020; Hu et al., 2022; Li et al., 2022; Xu et al., 2022). Thus, an important question that arises from previous work is: How can one design the *optimal* adaptation strategy? In this paper, we take a step towards answering this question.

Fine-tuning approaches to few-shot adaptation must manage a trade-off between adapting a large or small number of parameters. The former allows for better adaptation, but risks overfitting on a few-shot training set. The latter reduces the risk of overfitting, but limits the capacity for adaptation to novel categories and domains. The recent PMF (Hu et al., 2022) manages this trade-off through careful tuning of learning rates while fine-tuning the entire feature extractor. TSA (Li et al., 2022) and ETT (Xu et al., 2022) manage it by freezing the feature extractor weights, and inserting some parameter-efficient adaptation modules, lightweight enough to be trained in a few-shot manner. FLUTE (Triantafillou et al., 2021) manages it through selective fine-tuning of a tiny set of FILM (Perez et al., 2018) parameters, while keeping most of them fixed. Despite this progress, the best way to manage the adaptation/generalisation trade-off in fine-tuning approaches to few-shot learning (FSL) is still an open question. For example, which layers should be fine-tuned? What kind of adapters should be inserted, and where? While PMF, TSA, ETT, FLUTE, and others provide some intuitive recommendations, we propose a more systematic approach to answer these questions.

In this paper, we advance the adaptation-based paradigm for FSL by developing a neural architecture search (NAS) algorithm to find the optimal adaptation architecture. Given an initial pre-trained feature extractor, our NAS determines the subset of the architecture that should be fine-tuned, as

well as the subset of layers where adaptation modules should be inserted. We draw inspiration from recent work in NAS (Cai et al., 2020; Chen et al., 2021; Chu et al., 2021; Guo et al., 2020; Zhang et al., 2022) that proposes revised versions of the stochastic Single-Path One-Shot (SPOS) (Guo et al., 2020) weight-sharing strategy. Specifically, given a pre-trained ResNet (He et al., 2016) or Vision Transformer (ViT) (Dosovitskiy et al., 2021), we consider a search space defined by the inclusion or non-inclusion of task-specific adapters per layer, and the freezing or fine-tuning of learnable parameters per layer. Based on this search space, we construct a supernet (Brock et al., 2018) that we train by sampling a random path in each forward pass (Guo et al., 2020). Our supernet architecture is illustrated schematically in Figure 1b, where the aforementioned decisions are drawn as decision nodes ($\diamond$), and possible paths are marked in dotted lines.

While supernet training remains somewhat similar to standard NAS, the subsequent search poses new challenges in the FSL setting. Specifically, as cross-domain FSL considers novel domains/datasets at test time, the mainstream NAS paradigm of searching for a single neural architecture (Cai et al., 2019; Li et al., 2020b; Liu et al., 2019; Wang et al., 2021) is sub-optimal, as diverse downstream datasets likely prefer different architectures. On the other hand, conducting full-blown NAS per few-shot episode is too slow and would likely overfit to the small support set. Motivated by these challenges, we propose a novel NAS algorithm that shortlists a small number of architecturally diverse configurations at training time, but defers the final selection until the dataset and episode are known at test time. We empirically show that this is not only computationally efficient, but also improves results noticeably, especially when only one domain is available at training time. We term our method Neural Fine-Tuning Search (NFTS). NFTS defines a search space that is relevant to both convolutional and transformers architectures, and the choice of which specific adapter modules to consider is a hyperparameter, rather than a hard constraint.

Our contributions are summarised as follows: (i) We provide the first systematic Auto-ML approach to finding the optimal adaptation strategy to trade off adaptation flexibility and overfitting in multi-domain FSL. (ii) Our novel NFTS algorithm automatically determines which layers should be frozen or adapted, and where new adaptation parameters should be inserted for best few-shot adaptation. (iii) We advance the state-of-the-art in the well-established and challenging Meta-Dataset (Triantafillou et al., 2020), and the more recent and diverse Meta-Album (Ullah et al., 2022) benchmarks.

## 2 Neural Fine-Tuning Search

### 2.1 Few-Shot Learning Background

Let $\mathcal{D} = \{\mathcal{D}_i\}_{i=1}^D$ be the set of $D$ classification domains, and $\bar{\mathcal{D}} = \{X, Y\} \in \mathcal{D}$ a task containing $n$ samples along with their designated true labels $\{X, Y\} = \{x_j, y_j\}_{j=1}^n$. Few-shot classification is defined as the problem of learning to correctly classify a query set $\mathcal{Q} = \{X_\mathcal{Q}, Y_\mathcal{Q}\} \sim \bar{\mathcal{D}}$ by training on a support set $\mathcal{S} = \{X_\mathcal{S}, Y_\mathcal{S}\} \sim \bar{\mathcal{D}}$ that contains very few examples. This can be achieved by finding the parameters $\theta$ of a classifier $f_\theta$ with the objective

$$\arg\max_\theta \prod_\mathcal{D} p(Y_\mathcal{Q}|f_\theta(\mathcal{S}, X_\mathcal{Q})). \tag{1}$$

In practice, if $\theta$ is randomly initialised and trained using stochastic gradient descent on a small support set $\mathcal{S}$, it will overfit and fail to generalise to $\mathcal{Q}$. To address this issue, one can exploit knowledge transfer from some seen classes to the novel classes. Formally, each domain $\bar{\mathcal{D}}$ is partitioned into two disjoint sets $\bar{\mathcal{D}}_{\text{train}}$ and $\bar{\mathcal{D}}_{\text{test}}$, which are commonly referred to as "meta-train" and "meta-test", respectively. The labels in these sets are also disjoint, i.e., $Y_{\text{train}} \cap Y_{\text{test}} = \emptyset$. In that case, $\theta$ is trained by maximising the objective in Eq. 1 using the meta-train set, but the overall objective is to perform adequately when transferring knowledge to meta-test.

The knowledge transferred from meta-train to meta-test can take various forms (Hospedales et al., 2022). As discussed earlier, we aim to generalise a family of few-shot methods (Hu et al., 2022; Li et al., 2022; Xu et al., 2022) where parameters $\theta$ are transferred before a subset of them $\phi \subset \theta$ are fine-tuned; and possibly extended by attaching additional "adapter" parameters $\alpha$ that are trained for the target task. For meta-test, Eq. 1 can therefore be rewritten as

$$\arg\max_{\alpha,\phi} \prod_{\mathcal{D}_{\text{test}}} p(Y_\mathcal{Q}|f_{\alpha,\phi}(\mathcal{S}, X_\mathcal{Q})), \tag{2}$$

Our focus is on finding the optimal adaptation strategy in terms of (i) the optimal subset of parameters $\phi \subset \theta$ that need to be fine-tuned, and (ii) the optimal task-specific parameters $\alpha$ to add.

## 2.2 DEFINING THE SEARCH SPACE

Let $g_{\phi_k}$ be the minimal unit for adaptation in an architecture. We consider these to be the repeated units in contemporary deep architectures, e.g., a convolutional layer in a ResNet, or a self-attention block in a ViT. If the feature extractor $f_\theta$ comprises of $K$ such units with learnable parameters $\phi_k$, then we denote $\theta = \bigcup_{k=1}^{K} \phi_k$, assuming all other parameters are kept fixed. For brevity in notation we will now omit the indices and refer to every such layer as $g_\phi$. Following the state-of-the-art (Hu et al., 2022; Li et al., 2022; Triantafillou et al., 2021; Xu et al., 2022), let us also assume that task-specific adaptation can be performed either by inserting additional adapter parameters $\alpha$ into $g_\phi$, or by fine-tuning the layer parameters $\phi$.

This allows us to define the search space as two independent binary decisions per layer: (i) The inclusion or exclusion of an adapter module attached to $g_\phi$, and (ii) the decision of whether to use the pre-trained parameters $\phi$, or replace them with their fine-tuned counterparts $\phi'$. The size of the search space is, therefore, $(2^2)^K = 4^K$. For ResNets, we use the proposed adaptation architecture of TSA (Li et al., 2022), where a residual adapter $h_\alpha$, parameterised by $\alpha$, is connected to $g_\phi$

$$g_{\phi,\phi',\alpha}(x) = g_{\phi,\phi'}(x) + h_\alpha(x), \tag{3}$$

where $x \in \mathbb{R}^{W,H,C}$. For ViTs, we use the proposed adaptation architecture of ETT (Xu et al., 2022), where a tuneable prefix is prepended to the multi-head self-attention module $A_{qkv}$, and a residual adapter is appended to both $A_{qkv}$ and the feed-forward module $z$ in each decoder block

$$g_{\phi,\phi',\alpha}(x) = z(A_{qkv}[q \; ; \; g_{\phi,\phi'}(x)] + h_{\alpha 1}) + h_{\alpha 2}, \tag{4}$$

where $x \in \mathbb{R}^D$ and $[\cdot \; ; \; \cdot]$ denotes the concatenation operation. Note that in the case of ViTs the adapter is not a function of the input features, but simply an added offset.

Irrespective of the architecture, every layer $g_{\phi,\phi',\alpha}$ is parameterised by three sets of parameters, $\phi$, $\phi'$, and $\alpha$, denoting the initial parameters, fine-tuned parameters and adapter parameters respectively. Consequently, when sampling a configuration (i.e., path) from that search space, every such layer can be sampled as one of the variants listed in Table 1.

## 2.3 TRAINING THE SUPERNET

Following SPOS (Guo et al., 2020), our search space is actualised in the form of a supernet $f_{\theta,\alpha,\phi'}$; a "super" architecture that contains all possible architectures derived from the decisions detailed in Section 2.2. It is parameterised by: (i) $\theta$, the frozen parameters from the backbone architecture $f_\theta$, (ii) $\alpha$, from the adapters $h_\alpha$, and (iii) $\phi'$, from the fine-tuned parameters per layer $g_{\phi,\phi',\alpha}$.

We use a prototypical loss $\mathcal{L}(f, S, Q)$ as the core objective during supernet training and the subsequent search and fine-tuning.

$$\mathcal{L}(f, \mathcal{S}, \mathcal{Q}) = \frac{1}{|\mathcal{Q}|} \sum_{i=1}^{|\mathcal{Q}|} \log \frac{e^{-d_{cos}(C_{\mathcal{Q}_i}, f(\mathcal{Q}_i))}}{\sum_{j=1}^{|C|} e^{-d_{cos}(C_j, f(\mathcal{Q}c_i))}}, \tag{5}$$

where $C_{\mathcal{Q}_i}$ denotes the embedding of the class centroid that corresponds to the true class of $\mathcal{Q}_i$, and $d_{cos}$ denotes the cosine distance. The class centroids $C$ are the mean embeddings of support examples that belong to the same class: $C_l = \frac{1}{|\mathcal{S}^{y=l}|} \sum_{i=1}^{|\mathcal{S}|} f(\mathcal{S}_i^{y=l})$.

| | $\phi$ , $-$ | $\phi$ , $\alpha$ | $\phi'$ , $-$ | $\phi'$ , $\alpha$ |
|---|---|---|---|---|
| ResNet | $g_\phi(x)$ | $g_\phi(x) + h_\alpha(x)$ | $g_{\phi'}(x)$ | $g_{\phi'}(x) + h_\alpha(x)$ |
| ViT | $z(A_{qkv}[q \; ; \; g_\phi(x)])$ | $z(A_{qkv}[q \; ; \; g_\phi(x)] + h_{\alpha 1}) + h_{\alpha 2}$ | $z(A_{qkv}[q \; ; \; g_{\phi'}(x)])$ | $z(A_{qkv}[q \; ; \; g_{\phi'}(x)] + h_{\alpha 1}) + h_{\alpha 2}$ |

Table 1: The search space, as described in Section 2.2. A layer $g_{\phi,\phi',\alpha}$ can be sampled in one of the following variants: (i) $\phi$: fixed pre-trained parameters, no adaptation, (ii) $\alpha$: fixed pre-trained parameters, with adaptation, (iii) $\phi'$: fine-tuned parameters, no adaptation, (iv) $\phi', \alpha$ fine-tuned-parameters, with adaptation.

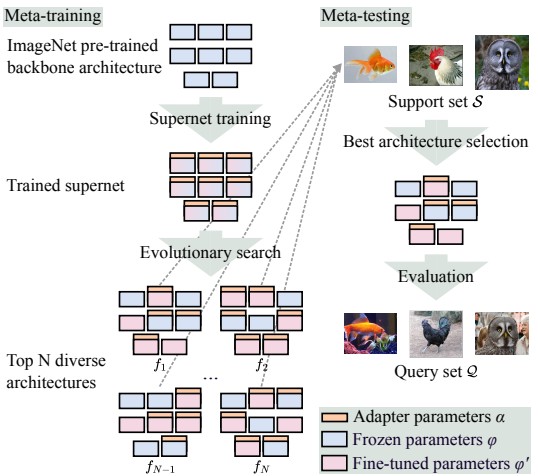

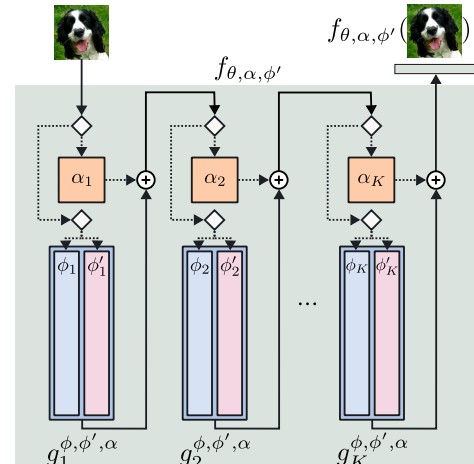

(a) After a supernet is trained, evolutionary search finds the top-performing candidates (validation set). During a new test episode, the shortlisted candidates are evaluated on the support set (Eq. 11), and the best architecture for that test episode is selected.

(b) The dotted lines represent possible paths that can be sampled during SPOS training. Every adaptable layer in the architecture ($g_i$) has its own pre-trained ($\phi_i \subset \theta$), fine-tuned ($\phi_i'$), and adapter ($\alpha_i$) parameters.

Figure 1: Our proposed NAS paradigm for few-shot adaptation. (a) Overall meta-train/meta-test workflow. (b) The supernet architecture. $f$ denotes the feature extractor, which is composed of many layers, $g$, which are the minimal unit for adaptation in our search space.

For supernet training, let $P$ be a set of size $4^K$, enumerating all possible sequences of $K$ layers that can be sampled from the search space. Denoting a path sampled from the supernet as $f_{\theta,\alpha,\phi'}^p$, we minimise the loss in Eq. 5 over multiple episodes and paths, so the final objective becomes:

$$\underset{\alpha,\phi'}{\arg\min}\ \mathbb{E}_{p\sim P}\mathbb{E}_{\mathcal{S},\mathcal{Q}}\ \mathcal{L}(f_{\theta,\alpha,\phi'}^p, \mathcal{S}, \mathcal{Q}). \tag{6}$$

In Appendix C, we summarise the supernet training algorithm in pseudocode (Algorithm 1).

## 2.4 Two-stage search for an optimal path

A supernet $f_{\theta,\alpha,\phi'}$ trained with the method described in Section 2.3 contains $4^K$ models, intertwined via weight sharing. As explained in Section 1, our goal is to search for the best-performing one, but the main challenge is related to the fact that we do not know what data is going to be used for adaptation at test time. One extreme approach would be to search for a single solution during training and simply use it throughout the entire test, regardless of the potential domain shift. Another, would be to defer the search and perform it from scratch each time a new support set is given to us at test time. However, both have their shortcomings. As such, we propose a hybrid, where searching is split into two phases – one during training, and a subsequent one during testing.

**Meta-training time** The search is responsible for pre-selecting a set of $N$ models from the entire search space. Its main purpose is to mitigate potential overfitting that can happen at test time, when only a small amount of data is available, while providing enough diversity to successfully adjust the architecture to the diverse set of test domains. Formally, we search for a sequence of paths $(p_1, p_2, ..., p_N)$ where:

$$p_k = \underset{p\in P}{\arg\max}\ \mathbb{E}_{\mathcal{S},\mathcal{Q}} A(f_{\theta,\alpha^*,\phi'^*}^p, \mathcal{S}, \mathcal{Q}), \quad \text{s.t.} \tag{7}$$

$$\alpha^*, \phi'^* = \underset{\alpha,\phi'}{\arg\min} \mathcal{L}(f_{\theta,\alpha,\phi'}^p, \mathcal{S}, \mathcal{S}) \tag{8}$$

$$\forall_{j=1,...,k-1}\ d_{cos}(p_k, p_j) \geq T, \tag{9}$$

where $T$ denotes a scalar threshold for the cosine distance between paths $p_k$ and $p_j$, and $A$ is the classification accuracy of a nearest centroid classifier (NCC) (Snell et al., 2017),

$$A(f, \mathcal{S}, \mathcal{Q}) = \frac{1}{|\mathcal{Q}|} \sum_{i=1}^{|\mathcal{Q}|} [\arg\min_j d_{cos}(C_{\mathcal{Q}_j}, f(\mathcal{Q}_i)) = Y_{\mathcal{Q}_i}]. \tag{10}$$

We measure accuracy of a solution using a query set, after fine-tuning on a separate support set (Eq. 8), then average across multiple episodes to avoid overfitting to a particular support set (Eq. 7). We also employ a diversity constraint, in the form of cosine distance between binary encodings of selected paths (Eq. 9), to allow for sufficient flexibility in the following test time search.

To efficiently obtain sequence $\{p_1, ..., p_N\}$, we use evolutionary search to find points that maximise Eq. 7, and afterwards select the $N$ best performers from the evolutionary search history that satisfy the constraint in Eq. 9. In Appendix C, we summarise the training-time search algorithm in pseudocode (Algorithm 2).

**Meta-testing time**  For a given meta-test episode, we decide which one of the pre-selected $N$ models is best-suited for adaptation on the given support set data. It acts as a failsafe to counteract the bias of the initial selection made at training time in cases when the support set might be particularly out-of-domain. Formally, the final path $p^*$ to be used in a particular episode is defined as:

$$p^* = \arg\min_{p \in \{p_1, ..., p_N\}} \mathcal{L}(f_{\theta, \alpha^*, \phi'^*}^p, \mathcal{S}, \mathcal{S}), \quad \text{s.t.} \tag{11}$$

$$\alpha^*, \phi'^* = \arg\min_{\alpha, \phi'} \mathcal{L}(f_{\theta, \alpha, \phi'}^p, \mathcal{S}, \mathcal{S}). \tag{12}$$

We test each of the $N$ models by fine-tuning it on the support set (Eq. 12) and scoring its loss on the same support set (Eq. 11). This is because the support set is the only source of data we have at test time and we cannot extract a disjoint validation set from it without risking the fine-tuning quality. It is important to note that, while this step risks overfitting, the pre-selection of models at training time, as described previously, should already limit the subsequent search to only models that are unlikely to overfit. Since $N$ is kept small in our experiments, we use a naive grid search to find $p^*$.

This approach is a generalization of the existing NAS approaches, as it recovers both when $N = 1$ or $N = 4^K$. Our claim is that intermediate values of $N$ are more likely to give us better results than any of the extremes, due to the reasons mentioned earlier. In particular, we would expect pre-selecting $1 < N \ll 4^K$ models to introduce reasonable overhead at test time while improving results, especially in cases when exposure to different domains might be limited at training time. In our evaluation we compare $N = 3$ and $N = 1$ to test this hypothesis. We do not include comparison to $N = 4^K$ as it is computationally infeasible in our setting (performing equivalent of training time search for each test episode would require us to fine-tune $\approx 14 \times 10^6$ models in total).

## 3 EXPERIMENTS

### 3.1 EXPERIMENTAL SETUP

**Evaluation on Meta-Dataset**  We evaluate NFTS on the extended version of Meta-Dataset (Requeima et al., 2019; Triantafillou et al., 2020), currently the most commonly used benchmark for few-shot classification, consisting of 13 publicly available datasets: FGVC Aircraft, CU Birds, Describable Textures (DTD), FGVCx Fungi, ImageNet, Omniglot, QuickDraw, VGG Flowers, CIFAR-10/100, MNIST, MSCOCO, and Traffic Signs. There are 2 evaluation protocols: single domain (SD) learning and multi-domain (MD) learning. In the single domain setting, only ImageNet is seen during training and meta-training, while in the multi-domain setting the first eight datasets are seen (FGVC Aircraft to VGG Flower). For meta-testing, 600 episodes are sampled for each domain, following the evaluation protocol proposed by Triantafillou et al. (2020).

**Evaluation on Meta-Album**  Further, we evaluate NFTS on the more recent Meta-Album (Ullah et al., 2022), which is more diverse than Meta-Dataset. We use the currently available Sets 0-2, which contain over 1000 unique labels across 30 datasets spanning 10 domains including microscopy, remote sensing, manufacturing, plant disease, character recognition, human action recognition tasks, etc. Unlike Meta-Dataset, in which the default evaluation protocol is variable-way

| | Method | Aircrafts | Birds | DTD | Fungi | ImageNet | Omniglot | QuickDraw | Flowers | CIFAR10 | CIFAR100 | MNIST | MSCOCO | Tr. Sign | Average |
|---|---|---|---|---|---|---|---|---|---|---|---|---|---|---|---|
| ResNet-18 | FLUTE (Liu et al., 2021a) | 48.5 | 47.9 | 63.8 | 31.8 | 46.9 | 61.6 | 57.5 | 80.1 | 65.4 | 52.7 | 80.8 | 41.4 | 46.5 | 52.6 |
| | ProtoNet (Snell et al., 2017) | 53.1 | 68.8 | 66.6 | 39.7 | 50.5 | 60.0 | 49.0 | 85.3 | - | - | - | 41.0 | 47.1 | 56.1 |
| | BOHB (Saikia et al., 2020) | 54.1 | 70.7 | 68.3 | 41.4 | 51.9 | 67.6 | 50.3 | 87.3 | - | - | - | 48.0 | 51.8 | 59.2 |
| | FO-MAML (Triantafillou et al., 2020) | 63.4 | 69.8 | 70.8 | 41.5 | 52.8 | 61.9 | 59.2 | 86.0 | - | - | - | 48.1 | 60.8 | 61.4 |
| | TSA (Li et al., 2022) | 72.2 | 74.9 | 77.3 | 44.7 | 59.5 | 78.2 | **67.6** | 90.9 | 82.1 | 70.7 | 93.9 | 59.0 | **82.5** | 73.3 |
| | NFTS | **74.9** | **76.5** | **81.6** | **50.5** | 62.7 | **80.2** | 67.2 | **94.5** | **83.0** | **71.5** | **94.0** | **59.7** | 81.9 | **75.2** |
| ViT-S | *PMF (Hu et al., 2022) | 76.8 | 85.0 | 86.6 | 54.8 | **74.7** | 80.7 | 71.3 | 94.6 | - | - | - | 62.6 | **88.3** | 77.5 |
| | ETT (Xu et al., 2022) | 79.9 | **85.9** | **87.6** | 61.8 | 67.4 | 78.1 | 71.3 | **96.6** | - | - | - | 62.3 | 85.1 | 77.6 |
| | NFTS | **83.0** | 85.5 | **87.6** | **62.2** | 71.0 | **81.9** | **74.5** | 96.0 | 79.4 | 72.6 | 95.2 | **62.6** | 87.9 | **79.2** |

Table 2: State-of-the art methods on Meta-Dataset. Single domain setting: only ImageNet is seen in meta-train. Mean acc. over 600 episodes. * Additional data used for training.

| | Method | Aircrafts | Birds | DTD | Fungi | ImageNet | Omniglot | QuickDraw | Flowers | CIFAR10 | CIFAR100 | MNIST | MSCOCO | Tr. Sign | Average |
|---|---|---|---|---|---|---|---|---|---|---|---|---|---|---|---|
| ResNet-18 | CNAPS (Requeima et al., 2019) | 83.7 | 73.6 | 59.5 | 50.2 | 50.8 | 91.7 | 74.7 | 88.9 | - | - | - | 39.4 | 56.5 | 66.9 |
| | Smpl. CNAPS (Bateni et al., 2020) | 82.0 | 74.8 | 68.8 | 46.6 | 58.4 | 91.6 | 76.5 | 90.5 | 74.9 | 61.3 | 94.6 | 48.9 | 57.2 | 69.5 |
| | Smpl. CNAPS [Ext.] (Bateni et al., 2022) | 84.1 | 76.8 | 69.0 | 48.8 | 58.8 | 93.9 | 78.6 | 91.6 | 75.7 | 62.9 | 95.7 | 48.7 | 76.1 | 73.9 |
| | SUR (Dvornik et al., 2020b) | 85.5 | 71.0 | 71.0 | 64.3 | 56.2 | 94.1 | 81.8 | 82.9 | 66.5 | 56.9 | 94.3 | 52.0 | 51.0 | 71.4 |
| | tri-M (Liu et al., 2021b) | 82.8 | 75.3 | 71.2 | 48.5 | 58.6 | 92.0 | 77.3 | 90.5 | 75.4 | 62.0 | 96.2 | 52.8 | 78.0 | 73.9 |
| | URT (Liu et al., 2021a) | 85.8 | 76.2 | 71.6 | 64.0 | 56.8 | 94.2 | 82.4 | 87.9 | 67.0 | 57.3 | 90.6 | 51.5 | 48.2 | 71.8 |
| | FLUTE (Triantafillou et al., 2021) | 82.8 | 75.3 | 71.2 | 48.5 | 58.6 | 92.0 | 77.3 | 90.5 | 75.4 | 62.0 | 96.2 | 52.8 | 63.0 | 72.7 |
| | URL (Li et al., 2021) | 89.4 | 80.7 | 77.2 | 68.1 | 58.8 | 94.5 | 82.5 | 92.0 | 74.2 | 63.5 | 94.7 | 57.3 | 63.3 | 76.6 |
| | TSA (Li et al., 2022) | 89.9 | 81.1 | 77.5 | 66.3 | 59.5 | 94.9 | 81.7 | 92.2 | 82.9 | 70.4 | 96.7 | 57.6 | 82.8 | 78.4 |
| | 2LM+TSA (Qin et al., 2023) | 89.3 | 82.1 | 78.2 | **69.5** | 58.4 | 95.4 | **82.8** | 92.4 | 76.5 | 67.7 | 97.3 | 57.3 | **88.4** | 79.5 |
| | SSA+TSA (Sreenivas & Biswas, 2023) | 90.0 | 82.2 | 77.6 | 66.6 | 58.9 | **95.6** | 82.7 | **93.0** | 82.9 | 70.8 | **98.5** | 58.1 | 84.9 | 80.1 |
| | NFTS | **90.1** | **83.8** | **82.3** | 68.4 | **61.4** | 95.4 | 82.6 | 92.2 | **83.0** | **75.1** | 95.4 | **58.8** | 81.9 | **80.7** |
| ViT-S | †CTX (Doersch et al., 2020) | 79.4 | 80.6 | 75.5 | 51.5 | 62.7 | 82.2 | 72.7 | 95.3 | - | - | - | 59.9 | 82.6 | 74.2 |
| | *PMF (Hu et al., 2022) | 88.3 | 91.0 | **86.6** | 74.2 | **74.6** | 91.8 | 79.2 | **94.1** | - | - | - | 62.6 | **88.9** | 83.1 |
| | NFTS | **89.1** | **92.5** | 86.3 | **75.1** | **74.6** | **92.0** | **80.6** | 93.5 | 75.9 | 70.8 | 91.3 | **62.8** | 87.2 | **83.4** |

Table 3: State-of-the art methods on Meta-Dataset. Multi-domain setting: the first 8 datasets are seen in meta-train. Mean acc. over 600 episodes. † Different backbone. * Additional training data.

variable-shot, Meta-Album evaluation follows a 5-way variable-shot setting, where the number of shots is typically 1, 5, 10 and 20. For meta-testing, results are averaged over 1800 episodes.

**Architectures** We employ two different backbone architectures, a ResNet-18 (He et al., 2016) and a ViT-small (Dosovitskiy et al., 2021). Following TSA (Li et al., 2022), the ResNet-18 backbone is pre-trained on the seen domains with the knowledge-distillation method URL (Li et al., 2021) and, following ETT (Xu et al., 2022), the ViT-small backbone is pre-trained on the seen portion of ImageNet with the self-supervised method DINO (Caron et al., 2021). We consider TSA residual adapters (Li et al., 2022; Rebuffi et al., 2017) for ResNet and Prefix Tuning (Li & Liang, 2021; Xu et al., 2022) adapters for ViT. This is mainly to enable direct comparison with prior work on the same base architectures that use exactly these same adapter families, without introducing new confounders in terms of mixing adapter types (Li et al., 2022; Xu et al., 2022). However, our framework is flexible, meaning it can accept any adapter type, or even multiple types in its search space.

## 3.2 COMPARISON TO THE STATE-OF-THE-ART

**Meta-Dataset** The results on Meta-Dataset are shown in Table 2 and Table 3 for the single domain and multi-domain training settings respectively. We can see that NFTS obtains the best average performance across all the competitor methods for both ResNet and ViT architectures. The margins over prior state-of-the-art are often substantial for this benchmark with +1.9% over TSA in ResNet-18 single domain, +2.3% in multi-domain and +1.6% over ETT in ViT-small single domain. The increased margin in the multi-domain case is intuitive, as our framework has more data with which to learn the optimal path(s).

We re-iterate that PMF, ETT, and TSA are special cases of our search space corresponding respectively to: (i) Fine-tune all and include no adapters, (ii) Include ETT adapters at every layer while freezing all backbone weights, and (iii) Include TSA adapters at every layer while freezing all backbone weights. We also share initial pre-trained backbones with ETT and TSA (but not PMF, as they use a stronger pre-trained model with additional data). Thus, the margins achieved over these competitors are attributable to our systematic approach to finding suitable architectures, in terms of where to fine-tune and where to insert new adapter parameters.

**Meta-Album** The results on Meta-Album are shown in Table 4 as a function of number of shots within the 5-way setting, following Ullah et al. (2022). We can see that across the whole range of support set sizes, our NFTS dominates all of the well-tuned baselines from Ullah et al. (2022). The margins are substantial, greater than 5% at 5-way/5-shot operating point, for example. This result confirms that our framework scales to even more diverse datasets and domains than those considered previously in Meta-Dataset.

| | From Scratch | Fine Tuning | Matching Net | ProtoNet | FO-MAML | NFTS |
|---|---|---|---|---|---|---|
| 1-shot | 30.42 | 40.43 | 34.49 | 38.07 | 33.94 | **43.76** |
| 5-shot | 38.31 | 50.87 | 44.32 | 51.17 | 44.50 | **57.59** |
| 10-shot | 39.58 | 53.42 | 49.23 | 55.18 | 48.62 | **60.10** |
| 20-shot | 39.83 | 55.12 | 52.99 | 59.67 | 51.35 | **60.97** |

Table 4: Comparison of our method against Meta-Album baselines, as reported in Fig. 2b of their paper (Ullah et al., 2022). The setting is cross-domain 5-way [1, 5, 10, 20]-shot, and accuracy scores are averaged over 1800 tasks drawn from Set0, Set1 and Set2.

| | ResNet-18 | | ViT-S | |
|---|---|---|---|---|
| Method | SD | MD | SD | MD |
| $\phi$ , − | 67.8 | 67.8 | 71.8 | 71.8 |
| $\phi$ , $\alpha$ | 70.4 | 76.5 | 73.8 | 77.3 |
| $\phi'$ , − | 70.2 | 76.3 | 74.0 | 77.5 |
| $\phi'$ , $\alpha$ | **70.8** | **76.9** | **74.4** | **78.9** |
| NFTS-1 | 73.6 | 80.1 | 78.7 | 83.1 |
| NFTS-N | **75.2** | **80.7** | **79.2** | **83.4** |

| | N=1 | N=3 | N=10 | N=100 |
|---|---|---|---|---|
| Test acc. (Support) | 96.5 | 97.1 | 99.9 | 99.8 |
| Test acc. (Query) | 72.8 | **72.9** | 71.5 | 71.4 |
| S/Q Pearson $r$ | 0.31 | **0.35** | 0.28 | 0.18 |

Table 5: Ablation study on Meta-Dataset. **Left:** Comparing four special cases of the search space in terms of average accuracy: (i) $\phi, -$: No adaptation, no fine-tuning, (ii) $\phi, \alpha$: Adapt all, (iii) $\phi', -$: Fine-tune all, (iv) $\phi', \alpha$: Adapt and fine-tune all. NFTS-$\{1,N\}$ refer to conventional and deferred episode-wise NAS respectively. Mean accuracy over 600 test episodes. **Right:** Impact of the number of candidates $N$ in the deferred NAS shortlist. Mean accuracy over 30 test episodes.

## 3.3 FURTHER ANALYSIS

**Ablation Study** To analyse the role that our architecture search plays in few-shot performance more precisely, we also conduct an ablation study of our final model against four corners of our search space: (i) Initial model only, using a pre-trained feature extractor and simple NCC classifier, which loosely corresponds to SimpleShot (Wang et al., 2019), (ii) Full adaptation only, using a fixed feature extractor, which loosely corresponds to TSA (Li et al., 2022), ETT (Xu et al., 2022), FLUTE (Triantafillou et al., 2021), and others – depending on base architecture and choice of adapter, (iii) Fully fine-tuned model, which loosely corresponds to PMF (Hu et al., 2022), and (iv) Combination of full fine-tuning and adaptation. Table 5 (left) shows that full adaptation, full fine-tuning, and their combination, give improvement on the linear readout baseline. Our work is the first to note the better performance of combining fine-tuning and adapters ($\phi', \alpha$).

Most importantly, both variants of our NFTS substantially outperform the baselines, demonstrating the value of our overall framework compared to mainstream fixed fine-tuning patterns. Next, we can compare our top-1 adaptation architecture selection strategy against our progressive approach that defers the final architecture selection to an episode-wise decision (i.e., NFTS-1 vs. NFTS-N). Our deferred architecture selection improves upon fixing the top-1 architecture from meta-train, especially for the single source domain case. This is intuitive, because NAS on a single source domain (cf., multi-domain) condition is most at risk of over-tuning to that domain, and should benefit the most from learning and transferring a diverse pool of architectures to the target domains.

**Discovered Architectures** We next analyse: *What kind of adaptation architecture is discovered by our NAS strategy?* We first summarise results of the entire search space in terms of which layers are preferential to fine-tune or not, and which layers are preferential to insert adapters or not in Figure 2a. The blocks indicate layers (columns) and adapters/fine-tuning (rows), with the color indicating whether that architectural decision was positively (green) or negatively (red) correlated with validation performance. We can see that the result is complex, without a simple pattern, as assumed by existing work (Hu et al., 2022; Li et al., 2022; Xu et al., 2022). That said, our NAS does discover some interpretable trends. For example, adapters should be included at early/late ResNet-18 layers and not at layers 5-9.

We next show the top three performing paths subject to diversity constraint in Figure 2b. We see that these follow the strong trends in the search space from Figure 2a. For example, they always adapt ($\alpha$)

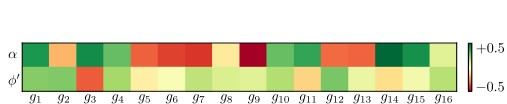

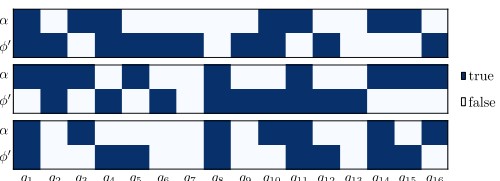

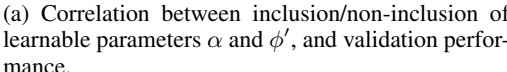

(a) Correlation between inclusion/non-inclusion of learnable parameters $\alpha$ and $\phi'$, and validation performance.

(b) Top 3 performing paths subject to diversity constraint. We expect the green areas in Fig. 2a to be roughly followed.

Figure 2: Qualitative analysis of our architecture search. Fig. 2a summarises the whole search space by answering the question: *How important is to adapt ($\alpha$) or fine-tune ($\phi'$) each block?* The color of each square indicates the point-biserial correlation (over all searched architectures) between adapting/fine-tuning layer $g_i$ and validation performance. Fig. 2b shows the top 3 performing candidates subject to a diversity constraint, after 15 generations of evolutionary search. Dark blue indicates that the layer is adapted/fine-tuned and light blue that it is not.

| | | | |
|---|---|---|---|
| CIFAR10 | 82.0 | 81.2 | **83.3** |
| CIFAR100 | **75.9** | 75.0 | 75.1 |
| MNIST | **95.5** | 94.4 | 95.1 |
| MSCOCO | **58.1** | 57.8 | 56.4 |
| Tr. Signs | 81.7 | **82.2** | 81.8 |

Table 6: How the diverse selection of architectures from Fig. 2b perform per unseen downstream domain in Meta-Dataset. Shading indicates episode-wise architecture selection frequency, numbers indicate accuracy using the corresponding architecture. The best dataset-wise architecture (bold) is most often selected (shading).

block 14 and never adapt block 9. However, due to our diversity constraint, they do include diverse decisions (e.g., whether to fine-tune ($\phi'$) block 15) that were not strongly indicated in Figure 2a.

Finally, we analyse how our small set of $N = 3$ candidate architectures in Figure 2b is used during meta-test. Recall that this small set allows us to perform an efficient minimal episode-wise NAS, including for novel datasets unseen during training. The results in Table 6 show how often each architecture is selected by held-out datasets during meta-test (shading), and what is the per-dataset performance using only that architecture. It shows how our approach successfully learns to select the most suitable architecture on a per-dataset basis, even for unseen datasets. This unique capability goes beyond prior work (Hu et al., 2022; Li et al., 2022; Xu et al., 2022) where all domains must rely on the same adaptation strategy despite their diverse adaptation needs.

**Why N=3? Impact of Candidate Architecture Set Size** Our framework admits various design options from $N = 1$, to large $N$ (full-blown NAS per few-shot episode). As discussed earlier, $N = 1$ uses the same architecture for all episodes without dataset-specific selection. Meanwhile, we expect large $N$ to suffer overfitting due to ultimately selecting a large number of parameters ($N$ networks, times the number of learnable parameters each) based on a small support set, defeating the purpose of our whole selective fine-tuning paradigm. To illustrate this we conduct a small experiment on a subset of 30 episodes[1] in Table 9 (right), comparing the support/train and query/test accuracy as a function of $N$. For large $N$, besides being expensive, we see overfitting with support accuracy approaching perfect, but query accuracy decreasing. This is also reflected in the decreasing Pearson correlation between episode-wise support and query accuracies as $N$ becomes large.

**Discussion** Overall, our deferred NAS approach where a large-scale search is conducted up-front during meta-train and a small candidate set search is conducted per meta-test episode, provides a reasonable trade-off between per-episode cost and efficacy. While our cost at recommended $N = 3$ is slightly higher than competitors with a single fine-tuning, it is similar or less than competitors who repeat adaptation with different learning rates during testing (Hu et al., 2022) ($4\times$ cost), or exploit a backbone ensemble ($8\times$ cost) (Dvornik et al., 2020b; Liu et al., 2021a). Where this cost is not acceptable, our single architecture NFTS-1 already provides state-of-the-art results.

---

[1]Selecting a high number of episodes is expensive, due to substantial per-episode architecture selection cost.

## 4 RELATED WORK

**Gradient-Based Few-Shot Adaptation**  Parameter-efficient adaptation modules have been applied for multi-domain learning, and transfer learning.  A seminal example is the Residual Adapter (Rebuffi et al., 2017), a lightweight 1x1 convolutional filter added to ResNet blocks. They were initially proposed for multi-domain learning, but were successfully used to achieve state of the art results for CNNs on the meta-dataset benchmark  (Triantafillou et al., 2020) by enabling fine-tuning of a URL (Li et al., 2021) pre-trained backbone without severe overfitting in the few-shot regime Li et al. (2022). Meanwhile, prompt (Jia et al., 2022) and prefix (Li & Liang, 2021) tuning are established examples of parameter-efficient adaptation for transformer architectures for similar reasons.  In FSL, Efficient Transformer Tuning (ETT) (Xu et al., 2022) apply a similar strategy to few-shot ViT adaptation using a DINO (Caron et al., 2021) pre-trained backbone.

PMF (Hu et al., 2022), FLUTE (Triantafillou et al., 2021) and FT (Dhillon et al., 2020) focus on adapting existing parameters without inserting new ones. To manage the adaptation/overfitting trade-off in the few-shot regime, PMF fine-tunes the whole ResNet or ViT backbone, but with carefully-managed learning rates. Meanwhile, FLUTE hand-picks a set of FILM parameters with a modified ResNet backbone for few-shot fine-tuning, while keeping the majority of the feature extractor frozen.

All of the methods above make heuristic choices about where to place adapters within the backbone, or for which parameters to allow/disallow fine-tuning. However, as different input layers represent different features (Chen et al., 2021; Zeiler & Fergus, 2014), there is scope for making better decisions about which features to update. Furthermore, in the multi-domain setting different target datasets may benefit from different choices about which modules to update. This paper takes an Auto-ML NAS-based approach to systematically address this issue.

**Neural Architecture search**  Neural Architecture Search (NAS) is a large topic (Elsken et al., 2019) which we do not attempt to review in detail here.  Mainstream NAS aims to discover new architectures that achieve high performance when training on a single dataset from scratch in a many-shot regime. To this end, research aims to develop faster search algorithms (Abdelfattah et al., 2021; Guo et al., 2020; Liu et al., 2019; Xiang et al., 2023), and better search spaces (Ci et al., 2021; Fang et al., 2020; Radosavovic et al., 2019; Zhou et al., 2021).  We build upon the popular SPOS (Guo et al., 2020) family of search strategies that encapsulate the entire search space inside a supernet that is trained by sampling paths randomly, and a search algorithm then determines the optimal path. We develop an instantiation of the SPOS strategy for multi-domain FSL. We construct a search space suited for parameter-efficient adaptation of a prior architecture to a new set of categories, and extend SPOS to learn on a suite of datasets, and efficiently generalise to novel datasets. This is different than the traditional SPOS paradigm of training and evaluating on the same dataset and label-space.

While there exist some recent NAS works that try to address a similar "train once, search many times" problem efficiently (Cai et al., 2020; Li et al., 2020a; Molchanov et al., 2022; Moons et al., 2021), naively using these approaches has two serious shortcomings: i) They assume that after the initial supernet training, subsequent searches do not involve any training (e.g., a search is only performed to consider a different FLOPs constraint while accuracy of different configurations is assumed to stay the same) and thus can be done efficiently – this is not true in the FSL setting as explained earlier.  ii) Even if naively searching for each dataset at test time were computationally feasible, the few-shot nature of our setting poses a significant risk of overfitting the architecture to the small support set considered in each episode.

## 5 CONCLUSIONS

In this paper we present NFTS, a novel framework for discovering the optimal adaptation architecture for gradient-based few-shot learning. NFTS contains several recent strong heuristic adaptation architectures as special cases within its search space, and we show that by systematic architecture search they are all outperformed, leading to a new state-of-the-art on Meta-Dataset and Meta-Album. While in this paper we use a simple and coarse search space for easy and direct comparison to prior work's hand-designed adaptation strategies, in future work we will extend this framework to include a richer range of adaptation strategies, and a finer-granularity of search.

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

## A ADDITIONAL RELATED WORK

**Feed-Forward Few-Shot Adaptation** Besides the gradient-based few-shot adaptation methods mentioned in Section 4, an alternative line of work (Requeima et al., 2019; Bateni et al., 2020) uses feed-forward networks to modulate the feature extraction process. However, these dynamic feature extractors are less able to generalise to completely novel domains than gradient-based methods (Finn & Levine, 2018), as the adaptation module itself suffers from an out of distribution problem.

**Sparse Fine-Tuning** In the many-shot regime there exists a line of work on sparse-fine tuning that attempts to identify a subset of parameters to update for fine-tuning, such as SpotTune (Guo et al., 2019) and DiffPrune (Guo et al., 2021). This is related to our goal in terms of working with sparse masks, but otherwise are completely different. They optimise for a sparse mask that optimises a single-task training loss on a large dataset, while we optimise for a small set of architectures that, in expectation, enables the learner to generalise to the test/query set of some novel after learning on a small train/support set in a few-shot episode.

**NAS and Few-Shot Learning** There have been a few prior attempts at NAS-related few-shot learning methods such as MetaNAS (Elsken et al., 2020) and mNAS (Wang et al., 2020a). These approaches differ from ours in that they attempt to learn architectures that are suited for learning from scratch, rather than how to fine-tune a powerful pre-trained feature; and have only been demonstrated on small architectures using a (flawed (Wang et al., 2021)) DARTS-like search, while lacking the ability to scale to the large architectures we consider. EG: Elsken et al. (2020) reports 7 GPU-days to search a $< 30k$ parameter model, while we take 2 GPU-days for 11M (ResNet-18) model.

## B ADDITIONAL RESULTS

### B.1 META-TRAINING CONVERGENCE

**Meta-Training** Fig. 3 shows convergence in terms of training and validation accuracy during supernet training (accuracy is averaged across episodes and paths, following the training objective in Eq. 8). This convergence curve corresponds to training a supernet with a ResNet-18 backbone in the multi-domain setting, for 40 thousand episodes. Similar behaviour occurs during the meta-training time search. The total cost for ResNet-18 is approximately 10 GPUh for supernet training, 30 GPUh searching, 0.05 GPUh per test episode.

**Path Search Process** In addition, we illustrate the path search process in Figure 4. This figure shows a 2D t-SNE projection of our $2K$-dimensional architecture search space, where the dots are candidate architectures of the evolutionary search process at different iterations. The dots are colored according to their validation accuracy. From the results we can see that: The initial set of candidates is broadly dispersed and generally low performing (left), and gradually converge toward a tighter cluster of high performing candidates (right). The top 3 performing paths subject to a diversity constraint (also illustrated in Fig. 2b) are annotated in purple outline.

### B.2 ADDITIONAL ABLATIONS

**Why not ensemble?** Our framework involves performing an episode-wise selection from our short-list candidate set of $N$ architectures, where the scoring of each architecture involves fine-tuning that architecture. One might therefore reasonably ask how our strategy compares to ensembling the same set of $N$ fine-tuned models? Our guiding hypothesis is that different test-time episodes likely need different architectures/adaptation schemes. For example, in single-domain meta-dataset,

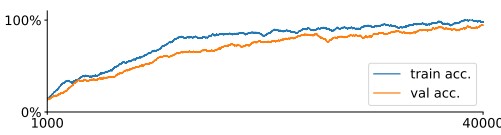

Figure 3: Meta-training convergence (40,000 train episodes).

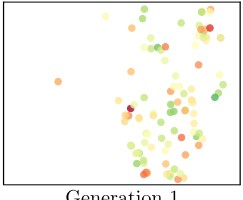 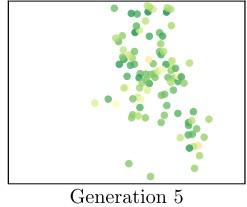 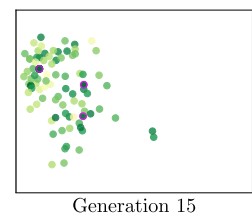 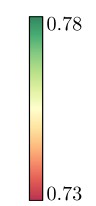

Generation 1    Generation 5    Generation 15

Figure 4: Population of paths(candidate architectures) in the search space after 1, 5, and 15 generations of evolutionary search. Each dot is a 2-d TSNE projection of the binary vector representing an architecture, and its color shows the validation performance for that architecture. The supernet contains a wide variety of models in terms of validation performance, and the search algorithm converges to a well-performing population. The top 3 performing paths that are given in Fig. 2b are highlighted in the far right figure (Generation 15) in purple outline.

|  | ResNet | ViT |
|---|---|---|
| Base Parameter Count | 11M | 22M |
| Number of Blocks | 16 | 12 |
| Parameter Count per Adapter | +36K-2M | +200K |
| Min-Max Possible Parameters Added in Search Space | 0 - 11M | 0 - 4M |
| Min Max Possible Parameters Updated in Search Space | 0 - 22M | 0 - 26M |
| Number of Actual Parameters after NAS (avg. over N=3 architectures) | 16M (+46%) | 25M (+14%) |
| Number of Updatable Parameters After NAS (avg. over N=3 architectures) | 15M | 24M |
| Actual Parameters Updated (TSA/ETT) | 11M | 4M |
| Actual Parameters Updated (PMF) | - | 22M |

Table 7: Architectural properties in the adaptation scheme of NFTS, for ResNet and ViT backbone architectures.

a downstream task more similar to ImageNet might prefer fewer updateable parameters compared to a downstream task more different to ImageNet. An architecture ensemble would take away this flexibility as the prediction architecture would always be a mixture of the $N$ architectures (even if they are episode-wise fine-tuned), rather than selecting the best suited architecture for each episode.

We validate our strategy with an ablation in Tab. 9 (ResNet-18, SD setting), showing that our second stage of architecture selection from the candidate short list outperforms simply predicting based on the ensemble of fine-tuned architectures.

**Detailed Results of Search Space Ablation** Tables 10 and 11 provide the exact scores per Meta-Dataset domain that are summarised in Table 5 of the main paper, for single domain and multi-domain FSL respectively.

**Structure and number of trainable parameters** We start with ResNet-18 and ViT, which are well-known architectures with well-documented number of parameters (11M and 22M respectively). These are fine-tuned and/or extended with additive modules: Between 0-16 TSA modules are added in the case of ResNet, and between 0-12 ETT modules are added in the case of VIT. The specific architecture of TSA and ETT modules given by Li et al. (2022) and Xu et al. (2022) respectively, and already summarised for completeness in Table 1 of our paper. Each TSA module has 100K-2M parameters (depending where in the architecture it is attached), and each ETT module has 200K parameters. The number of parameters increased by adapting is not a fixed quantity, it depends on the result of the architecture search (e.g.,: Fig 2b defines the result of the architecture search by showing which option in Table 1 is used at each layer). With respect to parameter counts of our search space, and of the final models discovered within that search space, we summarise them in Table 7 - together with a few related methods PMF, TSA, and ETT. Finally, we remark that although NFTS adds some parameters to the base network, our total number of parameters is still much less than the ensemble based methods like SUR (Dvornik et al., 2020b) and URT (Liu et al., 2021a), which are roughly 8x the base architecture size. Furthermore, in Table 8 we present a concise summary presenting parameter counts and computational costs between SUR, URL, and NFTS.

| Method | Backbone | # Parameters | Performance |
|---|---|---|---|
| SUR (Dvornik et al., 2020b) | ResNet-18 | 82M | 71.4 |
| URL (Li et al., 2021) | ResNet-18 | 82M | 76.6 |
| CTX (Doersch et al., 2020) | ResNet-34 | 21.4M | 74.2 |
| NFTS | ResNet-18 | 16M | 80.7 |

Table 8: Ablation study for comparison of number of parameters and performances between SUR, URL, and NFTS. Performance is mean accuracy scores Meta-Dataset, multi-domain setting.

| | NFTS-N (select best) | NFTS-N (ensemble) |
|---|---|---|
| Test acc. | 75.2 | 74.4 |

Table 9: Ablation: Selection vs ensemble (N=3, 600 test eps.).

## C  SUPERNET TRAINING AND PATH SEARCH ALGORITHMS

We present the detailed implementation (in pseudocode) for the supernet training and evolutionary search, in Algorithms 1 and 2, respectively.

## D  HYPERPARAMETER SETTING

Table 12 reports the hyperparameters used for all of our experiments. Note the following clarifications:

- "Number of epochs" refers to multiple forward passes of the same episode, while "Number of episodes" refers to the number of episodes sampled in total.
- The batch size is not mentioned, because we only conduct episodic learning, where we do not split the episode into batches, i.e., we feed the entire support and query set into our neural network architectures.
- Learning rate warmup, where applicable, occurs for the first 10% of the episodes.

We further specify something important: While our strongest competitors (Li et al., 2022; Xu et al., 2022) tune their learning rates for meta-testing (e.g., TSA uses LR=0.1 for seen domains and LR=1.0 for unseen, and ETT uses a different learning rate per downstream Meta-Dataset domain), we treat meta-testing episodes as completely unknown, and use the same hyperparameters we used on the validation set during search.

## E  SOURCE CODE

The source code is available at: `https://github.com/peustr/nfts-public`.

| | Method | Aircrafts | Birds | DTD | Fungi | ImageNet | Omniglot | QuickDraw | Flowers | CIFAR-10 | CIFAR-100 | MNIST | MSCOCO | Tr. Signs | Average |
|---|---|---|---|---|---|---|---|---|---|---|---|---|---|---|---|
| ResNet-18 | $\phi,-$ | 64.5 | 69.6 | 71.1 | 41.2 | 56.4 | 74.8 | 64.2 | 84.6 | 75.0 | 63.9 | 82.1 | 55.9 | 77.7 | 67.8 |
| | $\phi,\alpha$ | 69.6 | 67.7 | 75.0 | 42.5 | 59.5 | 71.3 | 64.9 | 88.8 | 77.4 | 70.0 | 90.2 | 58.4 | 80.1 | 70.4 |
| | $\phi',-$ | 69.9 | 74.7 | 73.3 | 39.5 | 57.3 | 71.9 | 65.4 | 89.0 | 76.5 | 66.3 | 93.6 | 54.4 | 81.4 | 70.2 |
| | $\phi',\alpha$ | 67.6 | 69.1 | 77.0 | 39.3 | 59.7 | 77.8 | 66.1 | 87.4 | 81.7 | 69.5 | 91.9 | 55.1 | 78.7 | 70.8 |
| | NFTS-1 | 73.2 | 76.5 | 81.6 | 42.1 | 61.3 | 80.2 | 66.9 | 90.0 | 82.9 | 68.8 | 94.0 | 58.4 | 80.6 | 73.6 |
| | NFTS-N | 74.9 | 76.5 | 81.6 | 50.5 | 62.7 | 80.2 | 67.2 | 94.5 | 83.0 | 71.5 | 94.0 | 59.7 | 81.9 | 75.2 |
| ViT-S | $\phi,-$ | 73.4 | 73.6 | 81.6 | 56.3 | 60.3 | 69.4 | 70.8 | 90.4 | 70.4 | 61.5 | 83.8 | 60.5 | 81.7 | 71.8 |
| | $\phi,\alpha$ | 76.9 | 83.2 | 86.7 | 59.3 | 63.7 | 75.8 | 65.1 | 89.5 | 70.7 | 67.4 | 81.1 | 54.8 | 82.9 | 73.8 |
| | $\phi',-$ | 76.8 | 80.9 | 85.8 | 61.4 | 65.9 | 73.2 | 68.5 | 91.0 | 69.9 | 66.1 | 82.5 | 57.6 | 78.8 | 74.0 |
| | $\phi',\alpha$ | 77.0 | 83.4 | 82.4 | 58.6 | 66.7 | 73.1 | 65.0 | 95.9 | 76.7 | 66.1 | 87.7 | 58.7 | 82.9 | 74.4 |
| | NFTS-1 | 83.0 | 85.5 | 87.3 | 62.2 | 68.8 | 81.9 | 72.9 | 95.3 | 79.4 | 72.6 | 95.2 | 62.6 | 87.5 | 78.7 |
| | NFTS-N | 83.0 | 85.5 | 87.6 | 62.2 | 71.0 | 81.9 | 74.5 | 96.0 | 79.4 | 72.6 | 95.2 | 62.6 | 87.9 | 79.2 |

Table 10: Ablation study on Meta-Dataset comparing four special cases of the search space: (i) $\phi,-$: No adaptation, no fine-tuning, (ii) $\phi,\alpha$: Adapt all, (iii) $\phi',-$: Fine-tune all, (iv) $\phi',\alpha$: Adapt and fine-tune all. NFTS-{1,N} refer to conventional and deferred episode-wise NAS respectively. Single domain setting: Only ImageNet is seen during training and search. Reporting mean accuracy over 600 episodes.

| | Method | Aircrafts | Birds | DTD | Fungi | ImageNet | Omniglot | QuickDraw | Flowers | CIFAR-10 | CIFAR-100 | MNIST | MSCOCO | Tr. Signs | Average |
|---|---|---|---|---|---|---|---|---|---|---|---|---|---|---|---|
| ResNet-18 | $\phi,-$ | 64.5 | 69.6 | 71.1 | 41.2 | 56.4 | 74.8 | 64.2 | 84.6 | 75.0 | 63.9 | 82.1 | 55.9 | 77.7 | 67.8 |
| | $\phi,\alpha$ | 89.3 | 78.3 | 76.1 | 62.7 | 57.2 | 93.8 | 76.0 | 90.8 | 77.8 | 66.1 | 90.5 | 56.9 | 79.5 | 76.5 |
| | $\phi',-$ | 90.2 | 76.7 | 70.6 | 63.1 | 57.8 | 88.2 | 79.3 | 88.9 | 78.2 | 68.2 | 96.1 | 51.7 | 82.9 | 76.3 |
| | $\phi',\alpha$ | 86.1 | 78.9 | 77.2 | 60.5 | 57.6 | 94.1 | 79.5 | 86.5 | 81.0 | 67.2 | 96.1 | 52.6 | 81.8 | 76.9 |
| | NFTS-1 | 90.1 | 82.1 | 79.9 | 67.9 | 61.4 | 94.3 | 82.6 | 92.2 | 82.4 | 73.8 | 95.4 | 58.1 | 81.0 | 80.1 |
| | NFTS-N | 90.1 | 83.8 | 82.3 | 68.4 | 61.4 | 94.3 | 82.6 | 92.2 | 83.0 | 75.1 | 95.4 | 58.8 | 81.9 | 80.7 |
| ViT-S | $\phi,-$ | 73.4 | 73.6 | 81.6 | 56.3 | 60.3 | 69.4 | 70.8 | 90.4 | 70.4 | 61.5 | 83.8 | 60.5 | 81.7 | 71.8 |
| | $\phi,\alpha$ | 85.7 | 84.3 | 81.8 | 68.7 | 70.4 | 89.1 | 77.0 | 90.2 | 73.5 | 61.4 | 82.6 | 53.7 | 72.4 | 77.3 |
| | $\phi',-$ | 83.0 | 84.5 | 81.1 | 70.9 | 72.4 | 88.6 | 74.6 | 90.4 | 75.1 | 63.5 | 87.0 | 54.0 | 75.5 | 77.5 |
| | $\phi',\alpha$ | 82.5 | 85.9 | 82.7 | 68.9 | 73.7 | 90.4 | 77.1 | 94.0 | 73.4 | 66.2 | 85.9 | 55.9 | 77.4 | 78.9 |
| | NFTS-1 | 89.1 | 90.3 | 86.3 | 75.1 | 74.6 | 92.0 | 80.6 | 93.5 | 75.9 | 70.8 | 91.3 | 62.7 | 87.2 | 83.1 |
| | NFTS-N | 89.1 | 92.5 | 86.3 | 75.1 | 74.6 | 92.0 | 80.6 | 93.5 | 75.9 | 70.8 | 91.3 | 62.8 | 87.2 | 83.4 |

Table 11: Ablation study on Meta-Dataset comparing four special cases of the search space: (i) $\phi,-$: No adaptation, no fine-tuning, (ii) $\phi,\alpha$: Adapt all, (iii) $\phi',-$: Fine-tune all, (iv) $\phi',\alpha$: Adapt and fine-tune all. NFTS-{1,N} refer to conventional and deferred episode-wise NAS respectively. Multi-domain setting: The first 8 datasets are seen during training and search. Reporting mean accuracy over 600 episodes.

---

**Algorithm 1:** Supernet training.

---

**Input:** Supernet $f_{\theta,\alpha,\phi'}$. Datasets $\mathcal{D}$. Step sizes $\eta_1$, $\eta_2$. Path pool $P$. Prototypical loss $\mathcal{L}$ (Eq. 5).

**Output:** Trained supernet $f_{\theta,\alpha,\phi'}$.

**repeat**

 Sample dataset $\bar{\mathcal{D}} \sim \mathcal{D}$

 Sample episode $\mathcal{S}, \mathcal{Q} \sim \bar{\mathcal{D}}$

 Sample path $p \sim P$ with learnable parameters $\alpha_p$, $\phi'_p$ and frozen parameters $\phi_p \subset \theta$

 $\alpha_p \longleftarrow \alpha_p - \eta_1 \nabla_{\alpha_p} \mathcal{L}(f^p_{\theta,\alpha,\phi'}, \mathcal{S}, \mathcal{Q})$

 $\phi'_p \longleftarrow \phi'_p - \eta_2 \nabla_{\phi'_p} \mathcal{L}(f^p_{\theta,\alpha,\phi'}, \mathcal{S}, \mathcal{Q})$

**until** prototypical loss converges

---

---

**Algorithm 2:** Training time evolutionary search.

---

**Input:** Supernet $f_{\theta,\alpha,\phi'}$. Datasets $\mathcal{D}$. Step sizes $\eta_1, \eta_2$. Prototypical loss $\mathcal{L}$ (Eq. 5). NCC
accuracy $A$ (Eq. 10).
**Output:** Optimal path $p^*$.
Initialise population $P$ randomly
Initialise fitness of $P$ as $\Psi_P \longleftarrow 0$
**repeat**
    Sample episodes from all datasets $\mathcal{S}, \mathcal{Q} \sim \mathcal{D}$
    **for** each candidate $p \in P$ **do**
        **for** a small number of epochs **do**
            $\alpha_p \longleftarrow \alpha_p - \eta_1 \nabla_{\alpha_p} \mathcal{L}(f^p_{\theta,\alpha,\phi'}, \mathcal{S}, \mathcal{S})$
            $\phi'_p \longleftarrow \phi'_p - \eta_2 \nabla_{\phi'_p} \mathcal{L}(f^p_{\theta,\alpha,\phi'}, \mathcal{S}, \mathcal{S})$
        **end**
        $\Psi_p \longleftarrow A(f^p_{\theta,\alpha,\phi'}, \mathcal{S}, \mathcal{Q})$
    **end**
    offspring $\longleftarrow$ recombine the $M$ best candidates of $P$ w.r.t. $\Psi_P$
    $P \longleftarrow P$ + offspring
    eliminate the $M$ worst candidates of $P$ wrt $\Psi_P$
**until** convergence or max. iterations

---

| | Hyperparameter | ResNet-18 | | | ViT-S | |
| | | SDL (MD) | MDL (MD) | MDL (MA) | SDL (MD) | MDL (MD) |
|---|---|---|---|---|---|---|
| | Backbone architecture | URL | URL | Supervised | DINO | DINO |
| | Adapter architecture | TSA | TSA | TSA | ETT | ETT |
| TRAIN | Number of episodes | 50000 | 80000 | 20000 | 80000 | 160000 |
| | Number of epochs | 1 | 1 | 1 | 1 | 1 |
| | Optimizer | adadelta | adadelta | adadelta | adamw | adamw |
| | Learning rate | 0.05 | 0.05 | 0.05 | 0.00007 | 0.00007 |
| | Learning rate schedule | - | - | - | cosine | cosine |
| | Learning rate warmup | - | - | - | linear | linear |
| | Weight decay | 0.0001 | 0.0001 | 0.0001 | 0.01 | 0.01 |
| | Weight decay schedule | - | - | - | cosine | cosine |
| SEARCH | Number of episodes | 100 | 100 | 100 | 100 | 100 |
| | Number of epochs | 20 | 20 | 20 | 40 | 40 |
| | Optimizer | adadelta | adadelta | adadelta | adamw | adamw |
| | Learning rate | 0.1 | 0.1 | 0.1 | 0.000003 | 0.000003 |
| | Weight decay | 0.0001 | 0.0001 | 0.0001 | 0.1 | 0.1 |
| | Initial population size | 64 | 64 | 64 | 64 | 64 |
| | Top-K crossover | 8 | 8 | 8 | 8 | 8 |
| | Mutation chance | 5% | 5% | 5% | 5% | 5% |
| | Top-N paths | 3 | 3 | 3 | 3 | 3 |
| | Diversity threshold | 0.4 | 0.4 | 0.4 | 0.2 | 0.2 |
| TEST | Number of episodes | 600 | 600 | 1800 | 600 | 600 |
| | Number of epochs | 40 | 40 | 40 | 40 | 40 |
| | Optimizer | adadelta | adadelta | adadelta | adamw | adamw |
| | Learning rate | 0.1 | 0.1 | 0.1 | 0.000003 | 0.000003 |
| | Weight decay | 0.0001 | 0.0001 | 0.0001 | 0.1 | 0.1 |
| | Regulariser strength | 0.04 | 0.04 | 0.04 | - | - |

Table 12: Hyperparameter setting for all experiments presented in Section 3 of the main paper. The notation is as follows: SDL=Single domain learning, MDL=Multi-domain learning, MD=Meta-Dataset, MA=Meta-Album, TRAIN=Supernet training phase, SEARCH=Evolutionary search phase, TEST=Meta-test phase.

