# OpenReview forum: "Neural Fine-Tuning Search for Few-Shot Learning"
_ICLR.cc/2024/Conference — ICLR 2024 oral_

### Official Review · Reviewer_pojm · 2023-10-29

**Soundness:** 3 good
**Presentation:** 2 fair
**Contribution:** 3 good
**Rating:** 6
**Confidence:** 4

**Summary:**

- The authors propose the optimal adaptation method through the lens of neural architecture search (NAS) in few-shot recognition.
- Given a pre-trained neural network, the proposed algorithm discovers the optimal arrangement of adapters, which layers to keep frozen, and which to fine-tune.
- The authors demonstrate the generality of our NAS method by applying it to both residual networks and vision transformers and report state-of-the-art performance on Meta-Dataset and Meta-Album.

**Strengths:**

(+) The proposed methods find some interpretable trends using layer-wise adaptations, which include the early/late layers of ResNet and ViT.

**Weaknesses:**

- (-) The authors stated the superior performances in various experimental settings. However, the author didn’t specify the structure and the number of parameters.
- (-) There is no ablation study on the two-stage search for optimal path (sec. 2.4): the best-performing path during training time, the searching path at test time, and the proposed hybrid one.

**Questions:**

- What is the most differentiating point from the prior NAS structures?
- How many parameters increased by adapting?
- Could authors provide parameter tables comparing NFTS (ResNet18) with others? The detailed architectural layout could be helpful to understand better.

**Details Of Ethics Concerns:**

None.

---

> ### Author Response · Authors · 2023-11-20
> **Response to Reviewer pojm**
>
> Thank you for your review. We understand that you would appreciate further details about the discovered architectures. We hope to address your concerns below.
> - **Structure and number of parameters:** We start with ResNet-18 and ViT, which are well-known architectures with well-documented number of parameters (11M and 22M respectively). These are fine-tuned and/or extended with additive modules: Between 0-16 TSA modules are added in the case of ResNet, and between 0-12 ETT modules  are added in the case of VIT. The specific architecture of TSA and ETT modules given in (Li et al 2022, Xu et al 2022) respectively, and already summarised for completeness in Table 1 of our paper. Each TSA module has 100K-2M parameters (depending where in the architecture it is attached), and each ETT module has 200K parameters. The number of parameters increased by adapting is not a fixed quantity, it depends on the result of the architecture search (e.g.,: Fig 2b defines the result of the architecture search by showing which option in Table 1 is used at each layer). With respect to parameter counts of our search space, and of the final models discovered within that search space, we summarise them below - together with a few related methods PMF, TSA, and ETT. Finally, we remark that although NFTS adds some parameters to the base network, our total number of parameters is still much less than the ensemble based methods like SUR (ECCV20) and URT (ICLR21), which are roughly 8x the base architecture size.
> | | ResNet-18 | ViT-S |
> | --- | :---: | :---: |
> | Base Parameter Count | 11M | 22M |
> | Number of Blocks | 16 | 12 |
> | Parameter Count per Adapter | +36K-2M | +200K |
> | Min-Max Possible Parameters Added in Search Space | 0 - 11M | 0 - 4M |
> | Min Max Possible Parameters Updated in Search Space | 0 - 22M | 0 - 26M |
> | Number of Actual Parameters after NAS (avg. over N=3 architectures) | 16M (+46%) | 25M (+14%) |
> | Number of Updatable Parameters After NAS (avg. over N=3 architectures) | 15M | 24M |
> | Actual Parameters Updated (TSA/ETT) | 11M | 4M |
> | Actual Parameters Updated (PMF) | - | 22M |
> - **The most differentiating point from the prior NAS structures:** Previous NAS methods focused on constructing architectures that are trained from scratch using large datasets. We develop a NAS approach specifically for searching how to adapt a pre-trained foundation model using a tiny few-shot dataset. The two crucial components of a NAS algorithm are 1) the search algorithm, and 2) the search space. We customize both of these for the few-shot learning problem. We design a search space that consists of the options to fine-tune or not each module (obviously fine-tuning is not applicable in standard NAS search spaces for models trained from scratch), and options to include additional adapters or not. For the search algorithm we extend SPOS into a two-stage algorithm that does most of the searching up-front during meta-train, and selects among a short list of candidate architectures during each few-shot learning episodes. This balances efficacy with runtime computational cost, as well as overfitting with underfitting. The positioning with respect to prior NAS is already explained in Related Work (Sec 4, paragraph “Neural Architecture Search”).
> - **Ablation study:** The requested ablation study is already covered by Tab 5. Please note that “N” in NFTS-N denotes how many candidate architectures are left to be selected at meta-test time.
>     - *1-stage vs 2-stage hybrid NAS:* This requested comparison corresponds to NFTS-1 vs NFTS-N in Tab 5 (left). NFTS-1 uses the best performing path identified during meta-training. Our proposed NFTS-N uses the two-stage/hybrid approach to select N candidate paths during meta-training, and then pick the best among these N (N=3) during meta-testing, which leads to outperforming NFTS-1 in Tab 5.
>     - *2-stage hybrid vs test-time NAS:* The third option mentioned by the reviewer (fully conducting NAS during each meta-testing episode) is prohibitively costly to run as it corresponds to  NFTS-N with N=2^K. Therefore, this cannot practically be ablated. The closest to this that we were practically able to run is NFTS-N with N=100, shown in Tab 5 (right). Already at N=100, where performance has degraded compared to N=3.
>     - The explanation for both of the above comparisons is given in Sec 3.3. To reiterate: Without any NAS at test-time, there is no opportunity to tune the architecture according to the downstream dataset (NFTS-N > NFTS-1). With too much NAS at test-time, there are too many free parameters to fit with the small few-shot support set, and it is possible to overfit the training set and perform badly on the support set (Tab 5, right).

---

> ### Comment · Reviewer_pojm · 2023-11-21
>
> Thank you for your detailed explanations and extensive experiments (clear ablation studies below). I increased my score from 5 to 6. I hope the comments (#parameters and performances) will be included in the future version of the paper.
>
> To be specific, one more ablation study (tables of #parameters and performances) could be prepared to compare the proposed method with others, including SUR(ECCV20) and URL(ICLR21).

---

> > ### Author Response · Authors · 2023-11-21
> > **Response to Reviewer pojm**
> >
> > Thank you for reading our feedback!
> > We believe that our response addresses your raised weaknesses (specific parameter counts, ablation study) and questions.
> > If you agree that they address the weaknesses, please consider raising your score.
> > If you have any outstanding concern, please let us know so that we can try to address them.

---

> ### Author Response · Authors · 2023-11-22
> **Response to Reviewer pojm**
>
> |Backbone | Method | Number of parameters | Performance on Meta-Dataset |
> | ----------- | ----------- | ----------- | ----------- |
> |ResNet18 | NFTS (Ours) | 16M | 80.7 |
> |ResNet18 | SUR (ECCV20) | ~88M | 71.4 |
> |ResNet18 | URL (ICLR21) | ~88M | 76.6 |
> |ResNet34 | CTX (NeurIPS20) | 21.4M | 74.2 |
>
>
> Thanks for the revised comments requesting another ablation study for comparison of #parameters and performances between SUR/URL and our method. From the table above, it is clearly seen that our method achieves better performance than several alternatives that have noticeably higher parameter counts. Please consider raising your score if you don't have further concerns.

---

### Official Review · Reviewer_eX3h · 2023-10-30

**Soundness:** 3 good
**Presentation:** 3 good
**Contribution:** 3 good
**Rating:** 8
**Confidence:** 3

**Summary:**

1. This paper provides the first systematic Auto-ML approach for finding the optimal adaptation strategy in few-shot learning.
2. This method designs a novel strategy for defining the search space.
3. The proposed method, namely NFTS, outperforms state-of-the-art methods in both Meta-Dataset and Meta-Album benchmarks.

**Strengths:**

1. The motivation for introducing NAS into FSL is good, as mentioned in this work: current FSL works have started to understand the trade-off between frozen weights and trained parameters. It makes sense to automatically search for the best configuration instead of manual search or "carefully tuning learning rates."

2. The experimental results present the superiority of NFTS; it achieves a significant performance gain on the Meta-Dataset.

3. The analysis is interesting as it shows the trend that the best-searched configuration does perform the best in the unseen downstream.

**Weaknesses:**

The results lack significance compared to the additional training required to obtain NFTs. The method requires training a supernet, performing an evaluation to find the best subnet. As NFTs achieve only a less than 1% accuracy gain on the Meta-Dataset in a multi-domain setting, the method is excessively computationally expensive and inefficient when compared to the actual performance gain.

**Questions:**

Could you offer some insights about their consistent adaptation of (α) block 14 and their lack of adaptation for block 9 in the 'Discovered Architectures' paragraph?

---

> ### Author Response · Authors · 2023-11-20
> **Response to Reviewer eX3h**
>
> Thank you for reviewing our paper, and for acknowledging the novelty of our method. Please, find our response to your concerns below.
> - **Computational cost vs. performance gain:** This is a very good point. We agree that MetaDataset is a very popular and highly competitive benchmark, where it is no longer possible to easily make gigantic leaps over the state-of-the-art. However, you raise a valid concern: “Is it worth it to perform an architecture search for 1-2% improvement?”. Our thinking of this is as follows: 1) The main cost of our method is the first stage SPOS architecture search, which is a one-off cost during meta-training. Since the results of this search can be re-used for many subsequent few-shot recognition tasks, it is not unreasonable to pay a large cost upfront. This rationale is used by many other methods with expensive upfront costs such as MAML and others that use expensive second-order gradients during meta-training. 2) Our architecture search cost is not actually “excessive” compared to alternatives. Our total meta-train cost is about 40GPUh, which is less than required for second order MAML on MetaDataset. It is also much less than methods that make heavy use of attention (e.g,: CrossTransformer requires 1300 V100 GPUh on MetaDataset - over 30x more than ours!). What is more, the majority of this cost comes from the meta-training-time search which can be trivially parallelized. 3) Our recurring meta-test cost (even including the small second-stage architecture search) is smaller than methods that use large feature extractor ensembles such as SUR (ECCV20) and URL (ICLR21), and/or cross-attention (URL, CTX). 4) A final benefit is that as a byproduct our method provides insight about your pre-trained architecture that may be valuable. E.g., as you point out in your question, some layers are always adapted, and some are never adapted.
> - **Architecture blocks consistently adapted/not adapted:** The vast majority of adaptation work hypothesizes that foundation models have a lot of knowledge encoded in their latent spaces. Having said that, allow us to answer your question with a toy example: Some architecture blocks may encode the notion of texture, or shape. Some other layers may encode the concept of a fish. The notion of texture is much more relevant to a large number of downstream visual recognition tasks than the concept of a fish, which is very domain-specific. This may be the reason why we see a “fish block” always adapted, but a “texture block” always in its original form, as it can already do its job pretty well. Of course, this is a hypothetical example and it is still an open research question to properly identify which neurons represent which concept, as would be required to interpret our specific discovered architectures in this way. But in the meanwhile, this is exactly why it is helpful to have automated methods to identify which blocks to adapt/not adapt.

---

> ### Comment · Reviewer_eX3h · 2023-11-21
>
> Thank you for your response. I will maintain my score. Given that computational cost is a core consideration in NAS-related work, including an analysis, such as the one you pointed out in your previous response, could further enhance the quality of your paper.

---

### Official Review · Reviewer_mc8g · 2023-11-01

**Soundness:** 4 excellent
**Presentation:** 4 excellent
**Contribution:** 4 excellent
**Rating:** 8
**Confidence:** 4

**Summary:**

The paper presents NFTS, a hierarchical method for neural architecture search in the few-shot image classification domain. The proposed framework engages various ways to adapt ResNet and ViT architectures to the support set including fine-tuning and adaptation parameters and then performs a search to identify the best-performing combination amongst each search path. The total number of paths is limited to both address computational limitations and prevent overfitting. Experiments on Meta-Dataset and Meta-Album demonstrate the strong efficacy of the approach when adapted inside a prototypical classifier with ResNet and ViT backbones. Ablation studies provide insights into various aspects of the method.

**Strengths:**

- The paper is very well-written.
- NFTS is empirically effective and demonstrates a good balance between enabling adaptation using the support set while preventing overfitting. Clear empirical evidence shows the model's ability to select a more optimal architectural combination than previous baselines.
- Experiments are extensively performed on large-scale datasets that demonstrate the efficacy of NFTS across both ResNet and ViTs.
- Ablation studies justify various architectural choices made such the total number of search paths and the granularity of options.

**Weaknesses:**

- Empirical results reported lack confidence intervals. Although I suspect this is due to space limitations, they should be included to verify the statistical significance of the results report and for better comparison with baselines. If some results do not meet statistical significance, they must be modified when presented in results tables to reflect so accordingly and the claims made need to be adjusted.
- Ablation study on search paths shows that N=3 not only provides better computational efficiency but additionally prevent overfitting on the support set. How does it compare with N=2 or N=4? I believe that further insights here would be useful as to how this hyperparameter is set. Furthermore, how does performance vary depending on N across ViT and ResNet?
- Meta-dataset baselines that are compared to omit some recent methods that can be included for completeness of comparison [1, 2, 3].

[1] Improved Few-Shot Visual Classification
[2] CrossTransformers: spatially-aware few-shot transfer
[3] Enhancing Few-Shot Image Classification with Unlabelled Examples
[4] Beyond Simple Meta-Learning: Multi-Purpose Models for Multi-Domain, Active and Continual Few-Shot Learning

**Questions:**

Please address the questions and limitations noted above. Overall, I believe that this is a strong submission, and the broader research community can benefit from it. I believe that the empirical results of the paper need to be verified in terms of statistical significance by providing the appropriate confidence intervals across the reported numbers. This is the only major weakness in the submission, and once addressed with the other limitations noted, I would be more than happy to recommend the paper for acceptance.

---

> ### Author Response · Authors · 2023-11-20
> **Response to Reviewer mc8g**
>
> Thank you for your review, we are happy that you found our paper well-written, strong, and benefitial to the community. Please, find our response to your concerns below.
> - **Confidence intervals:** Initially, we did not include the confidence intervals for two reasons: 1) They are all quite small; the majority of standard errors are < 0.1%. This should have been explicitly stated in the paper. 2) As you accurately pointed out, the space limitations; since the confidence intervals are not very informative, adding a “±0.002” to every column would make the tables even longer and unreadable. Given the small standard errors on the means, most of the differences between methods are statistically significant. We do, however, have the confidence intervals for our experiments, and are happy to include them in the supplementary material. For example, see the confidence intervals (SEM) reported below (RN18/ViT, MDL):
> | Dataset | ResNet-18 | ViT-S |
> | --- | :---: | :---: |
> | Aircrafts | 90.1±0.013 | 89.1±0.002 |
> | Birds | 83.8±0.058 | 92.5±0.034 |
> | DTD | 82.3±0.002 | 86.3±0.002 |
> | Fungi | 68.4±0.116 | 75.1±0.086 |
> | ImageNet | 61.4±0.145 | 74.6±0.088 |
> | Omniglot | 94.3±0.037 | 92.0±0.047 |
> | QuickDraw | 82.6±0.020 | 80.6±0.041 |
> | VGG Flowers | 92.2±0.086 | 93.5±0.090 |
> | CIFAR10 | 83.0±0.001 | 75.9±0.002 |
> | CIFAR100 | 75.1±0.002 | 70.8±0.001 |
> | MNIST | 95.4±0.001 | 91.3±0.001 |
> | MSCOCO | 58.8±0.130 | 62.8±0.063 |
> | Traffic Signs | 82.9±0.107 | 87.2±0.032 |
> - **Comparison N=2,4, ResNet/ViT:** In practice, N is a hyperparameter that is chosen heuristically, and the results do not change much when N=2,3,4,5, etc. The big difference comes from N=1, N=small, N=large. This is true for both ResNet and ViT architectures. Please, find an analysis of small Ns below (RN18, MDL, N=2,3,4):
> | | N=2 | N=3 | N=4 |
> | --- | :---: | :---: | :---: |
> | Mean acc. (MetaDataset, NumEps=600) | 80.3 | 80.7 | 80.7 |
> Since the differences between small values of N are insignificant, in our experiments we arbitrarily chose N=3 as a value that is greater than 1 but small enough not to impose a substantial computational burden. We did not attempt to carefully tune this hyperparameter.
> - **Additional baselines:** Thank you for providing these additional references, we have added them to Table 3.

---

### Public Comment · ~Chi_Zhang13 · 2024-09-11
**Relations to a piror work**

Dear Authors,

I had the opportunity to read your research, and I noticed that it shares conceptual similarities with an earlier work titled Meta Navigator [1], published at ICCV 2023. Specifically, both approaches explore the idea of using a search strategy to identify which layers need adaptation and the extent of that adaptation. Furthermore, both studies focus on few-shot learning. Unfortunately, I did not find any discussion or citation of this related work in your paper. I would suggest that you address this point.

[1] Meta Navigator: Search for a Good Adaptation Policy for Few-shot Learning

---

### Meta-Review · Area_Chair_owz4 · 2023-12-07

**Metareview:**

The submission presents a hierarchical approach to neural architecture search for few-shot image classification applications. The approach, called NTFS, aims to discover the optimal arrangement of adapters and configuration of frozen and fine-tuned layers. Experiments on Meta-Dataset and Meta-Album when applying NTFS on ResNet and ViT backbones demonstrate strong performance, and ablation studies further break down the influence of various design decisions.

Reviewers note the submission's writing quality (mc8g). They find the proposed approach to be sound (eX3h) and effective (mc8g eX3h), as demonstrated through extensive experiments (mc8g). All reviewers find the ablation studies to be insightful. Concerns over the lack of confidence intervals (mc8g), the computational cost vs performance tradeoff (eX3h), and missing comparisons and ablations (mc8g, pojm) have been addressed by the authors in their response.

All reviewers agree on acceptance. I encourage the authors to incorporate the discussion on computational cost and parameter counts in their final manuscript.

**Justification For Why Not Higher Score:**

N/A

**Justification For Why Not Lower Score:**

The submission received strong support from two out of three reviewers, and the proposed approach has a wide application potential that would be worth highlighting in an oral presentation.

---

### Decision · Program_Chairs · 2024-01-16

Accept (oral)